# The RFamide receptor DMSR-1 regulates stress-induced sleep in *C. elegans*

**Michael J Iannacone[1,2], Isabel Beets[3], Lindsey E Lopes[1,2], Matthew A Churgin[4], Christopher Fang-Yen[4], Matthew D Nelson[5], Liliane Schoofs[3], David M Raizen[1,2]***

[1]Department of Neurology, Perelman School of Medicine, University of Pennsylvania, Philadelphia, United States; [2]Center for Sleep and Circadian Neurobiology, Perelman School of Medicine, University of Pennsylvania, Philadelphia, United States; [3]Department of Biology, Katholieke Universiteit Leuven, Leuven, Belgium; [4]Department of Bioengineering, School of Engineering and Applied Sciences, University of Pennsylvania, Philadelphia, United States; [5]Department of Biology, Saint Joseph's University, Philadelphia, United States

**Abstract** In response to environments that cause cellular stress, animals engage in sleep behavior that facilitates recovery from the stress. In *Caenorhabditis elegans*, stress-induced sleep (SIS) is regulated by cytokine activation of the ALA neuron, which releases FLP-13 neuropeptides characterized by an amidated arginine-phenylalanine (RFamide) C-terminus motif. By performing an unbiased genetic screen for mutants that impair the somnogenic effects of FLP-13 neuropeptides, we identified the gene *dmsr-1*, which encodes a G-protein coupled receptor similar to an insect RFamide receptor. DMSR-1 is activated by FLP-13 peptides in cell culture, is required for SIS *in vivo*, is expressed non-synaptically in several wake-promoting neurons, and likely couples to a Gi/o heterotrimeric G-protein. Our data expand our understanding of how a single neuroendocrine cell coordinates an organism-wide behavioral response, and suggest that similar signaling principles may function in other organisms to regulate sleep during sickness.

**\*For correspondence:** raizen@
mail.med.upenn.edu

**Competing interests:** The authors declare that no competing interests exist.

## Introduction

When acutely ill, animals engage in a behavioral sequence that includes cessation of feeding and body movements as well as reduced responsiveness to the environment. During acute infectious illness in mammals, electrophysiological correlates of sleep behavior are observed (*Toth and Krueger, 1988*, *1989*), indicating that the behavioral sequence is sleep. In the arthropod *Drosophila melanogaster* (*Williams et al., 2007*; *Lenz et al., 2015*) and the nematode *Caenorhabditis elegans* (*Hill et al., 2014*), acute illness results in cellular stress, which then induces a sleep behavior: the animals stop moving and feeding, do not respond to weak stimuli but move normally in response to strong stimuli (*Hill et al., 2014*). This stress-induced sleep (SIS, also known as sickness sleep) is beneficial to the animal and helps it recover from the acute injury (*Spiegel et al., 2002*; *Hill et al., 2014*; *Kuo and Williams, 2014*; *Fry et al., 2016*). In fruit flies and round worms, environments that induce SIS include bacterial pathogens, bacterial toxins, heat shock, cold shock, osmotic shock, and ultraviolet light exposure (*Hill et al., 2014*; *Lenz et al., 2015*). A comparison between mammalian sickness sleep and invertebrate SIS was recently reviewed (*Davis and Raizen, 2016*).

Nematode SIS is a distinct sleep state from a larval sleep state known as developmentally timed sleep (DTS) (*Trojanowski et al., 2015*), which is regulated by a homolog of the core circadian protein PERIOD (*Monsalve et al., 2011*). In the absence of stress, nematodes experience sleep only when they transition between larval stages but do not sleep in the adult stage. Since *C. elegans*

**eLife digest** People often feel fatigued and sleepy when they are sick. Other animals also show signs of sleepiness when ill – they stop eating, move less, and are less responsive to changes in their environment. Sickness-induced sleep helps both people and other animals to recover, and many scientists believe that this type of sleep is different than nightly sleep.

Studies of sickness-induced sleep have made use of a simple worm with a simple nervous system. In this worm, a single nerve cell releases chemicals that cause the worm to fall asleep in response to illness. Animals exposed to one of these chemicals, called FLP-13, fall asleep even when they are not sick. As such, scientists would like to know which cells in the nervous system FLP-13 interacts with, what receptor the cells use to recognize this chemical, and whether it turns on cells that induce sleep or turns off the cells that cause wakefulness.

Now, Iannacone et al. show that FLP-13 likely causes sleep by turning down activity in the cells in the nervous system that promote wakefulness. The experiments sifted through genetic mutations to determine which ones cause the worms not to fall asleep when FLP-13 is released. This revealed that worms with a mutation that causes them to lack a receptor protein called DMSR-1 do not become sleepy in response to FLP-13. This suggests that DMSR-1 must be essential for FLP-13 to trigger sleep. About 10% of cells in the worm's nervous system have the DMSR-1 receptor. Some of these neurons tell the worm to move forward or to forage around for food. The experiments also showed that FLP-13 is probably not the only chemical that interacts with the DMSR-1 receptor, but the identities of these other chemicals remain unknown.

Additional experiments are now needed to determine if sickness-induced sleepiness in humans and other mammals is triggered by a similar mechanism. If it is, then drugs might be developed to treat people experiencing fatigue associated with sickness as well as other unexplained cases of fatigue.

does not have an identifiable circadian rhythm of sleep, adult nematodes are an ideal system to study SIS in the absence of the circadian and homeostatic effects of animals that require daily sleep.

The mechanism of SIS is poorly-understood, yet a few common themes have emerged from studies across phylogeny. The acute illness can occur outside of the brain yet affect behavior, suggesting that communication occurs between non-neural and neural tissues. Cytokine signaling is involved. For example, in mammals, the cytokines interleukin-1 beta and tumor necrosis factor alpha, whose levels increase during an infectious challenge, are each sufficient to induce sleep when injected into the brain (reviewed in (*Krueger, 2008*). In nematodes (*Van Buskirk and Sternberg, 2007*), arthropods (*Foltenyi et al., 2007*), and mammals (*Kushikata et al., 1998*; *Kramer et al., 2001*), signaling by epidermal growth factor (EGF) is sufficient to induce sleep behavior and, at least in nematodes, EGF signaling is necessary for SIS (*Hill et al., 2014*). These cytokines act on central nervous system (CNS) neurons, which then induce sleep.

In mammals, CNS neurons that regulate sleep reside in the hypothalamus (*Saper et al., 2005b*). In *C. elegans*, the target for EGF action is a single interneuron called ALA (*Van Buskirk and Sternberg, 2007*), whose developmental program has similarities to the developmental program of mammalian neuroendocrine cells (*Van Buskirk and Sternberg, 2010*). With EGF activation, ALA depolarizes to release neuropeptides encoded by the gene *flp-13* (FMRFamide-Like Peptide-13) to promote sleep (*Nelson et al., 2014*). FLP-13 peptides are characterized by an amidated Arginine-Phenylalanine (RFamide) motif at their C-termini. RFamide neuropeptides are involved in many physiological functions in both invertebrates (*López-Vera et al., 2008*; *Peymen et al., 2014*), and vertebrates (*Rőszer and Bánfalvi, 2012*; *Kim, 2016*). In fruit flies, several RFamide neuropeptides regulate sleep (*He et al., 2013*; *Shang et al., 2013*), including the RFamide neuropeptide FMRFamide, which regulates SIS (*Lenz et al., 2015*).

In this study, we focused on understanding the downstream mechanism of the sleep-promoting activity of FLP-13 RFamide peptides. Both locomotion quiescence and feeding quiescence induced by *flp-13* can be reversed by activation of motor neurons (*Trojanowski et al., 2015*), suggesting that *flp-13* mediates quiescence at the level of the nervous system. Furthermore, quiescence induced by

*flp-13* requires the G protein alpha subunit GOA-1 (*Trojanowski et al., 2015*), suggesting that these peptides signal through a G protein-coupled receptor (GPCR). There are more than 150 genes in the *C. elegans* genome predicted to encode neuropeptide receptor GPCRs (*Frooninckx et al., 2012*; *Hobert, 2013*). In prior detailed analysis of one of these GPCRs (*Nelson et al., 2015*), we showed that while FRPR-4 can be activated by FLP-13 peptides in cell-based assay, its genetic removal does not abrogate *flp-13* induced quiescence in vivo, suggesting that it is not the receptor mediating the quiescence-inducing effects of FLP-13 peptides in response to cellular stress. Since we had no strong *a priori* reason to implicate other specific GPCRs, we took a hypothesis-independent forward genetic screen approach to identify the FLP-13 receptor (*Yuan et al., 2015*). We here identify the GPCR DMSR-1 (<u>D</u>ro<u>M</u>yo<u>S</u>uppressin <u>R</u>eceptor related-1) as required for *flp-13* somnogenic effects. DMSR-1 is expressed in about one tenth of all *C. elegans* neurons and localizes diffusely to membranes. FLP-13 peptides can activate the receptor directly, indicating that DMSR-1 is likely an in vivo receptor for FLP-13 neuropeptides. Inhibition of neurons where *dmsr-1* is expressed enhances the effect of *flp-13*, suggesting that DMSR-1 transduces the FLP-13 signal by reducing activity of wake-promoting neurons.

## Results

### *dmsr-1* is required for *flp-13* induced sleep

Our goal was to characterize the downstream mechanism for the sleep-inducing FLP-13 neuropeptides. We induced sleep by over expressing the *flp-13* gene in somatic cells under the control of the heat-inducible promoter *hsp-16.2* (<u>h</u>eat-<u>s</u>hock <u>p</u>rotein-16.2). P*hsp-16.2:flp-13* transgenic animals are quiescent for body and pharyngeal movements two hours after exposure to a heat pulse to induce *flp-13* gene expression (*Nelson et al., 2014*). As previously described (*Yuan et al., 2015*) and illustrated in *Figure 1A*, we performed a forward mutagenesis screen to identify mutants with defective quiescence in response to *flp-13* overexpression. By performing genetic complementation tests, we determined that five of the identified mutants (*qn45*, *qn49*, *qn51*, *qn52* and *qn53*) were alleles of the same gene (*Yuan et al., 2015*). We found two additional alleles (*qn40* and *qn44*) in this gene by manually screening for mutants with defective feeding quiescence induced by overexpression of *flp-13*.

To identify the gene mutated to cause the defect in *flp-13* induced quiescence, we sequenced the genomes of six of the allelic mutants. Each of the six mutants had a mutation in *dmsr-1*, a gene predicted to encode a GPCR (*Figure 1B*). Four of the mutants (*qn44*, *qn49*, *qn51* and *qn53*) contained premature stop codons. One mutant (*qn45*) contained a complex rearrangement in exon 4 consisting of deletion of 332 nucleotides coupled with insertion of 54 nucleotides (*Figure 1—figure supplement 1*). This rearrangement results in a frame shift that eliminates the three transmembrane domains at the C-terminus of the protein. One mutant (*qn52*) contained an alanine to valine mutation in exon 3. We Sanger sequenced the *dmsr-1* gene in *qn40*, and found a premature stop mutation in exon 4 (*Figure 1B*). The six mutations which caused premature stops or a deletion are predicted to eliminate gene function. We do not know whether the missense mutation in *qn52* eliminates genes function; however, based on the phenotype and the relatively conservative alanine to valine mutation, the second extracellular loop where this mutation occurred likely plays an important role in the function of DMSR-1. We performed subsequent analyses using the *qn45* allele because it was a convincing null mutation and because the genotype could be easily determined using polymerase chain reaction (PCR).

DMSR-1 is one of 41 *C. elegans* proteins in the RFamide receptor family (*Frooninckx et al., 2012*) and is orthologous to *Drosophila* receptors for the neuropeptides dromyosuppressin and FMRFamide. Like FLP-13 peptides, Dromyosuppressin and FMRFamide are neuropeptides with an amidated Arginine-Phenylalanine (RFamide) motif at their C-terminus (*Nachman et al., 1993*). We grouped the *C. elegans* DMSR and other RFamide receptor proteins based on their sequence homology to the *Drosophila* myosuppressin receptors MsR1 and MsR2, which are known to respond to RFamide peptides (*Figure 1C*). Other RFamide receptors in *C. elegans* include FRPR (<u>FMR</u>Famide <u>P</u>eptide <u>R</u>eceptor family) proteins, and the neuropeptide receptor family (NPR), although these FRPR and NPR receptors are more closely related to the *Drosophila* FMRFaR than they are to the myosuppressin related receptors. We included in the phylogenetic tree other identified RFamide

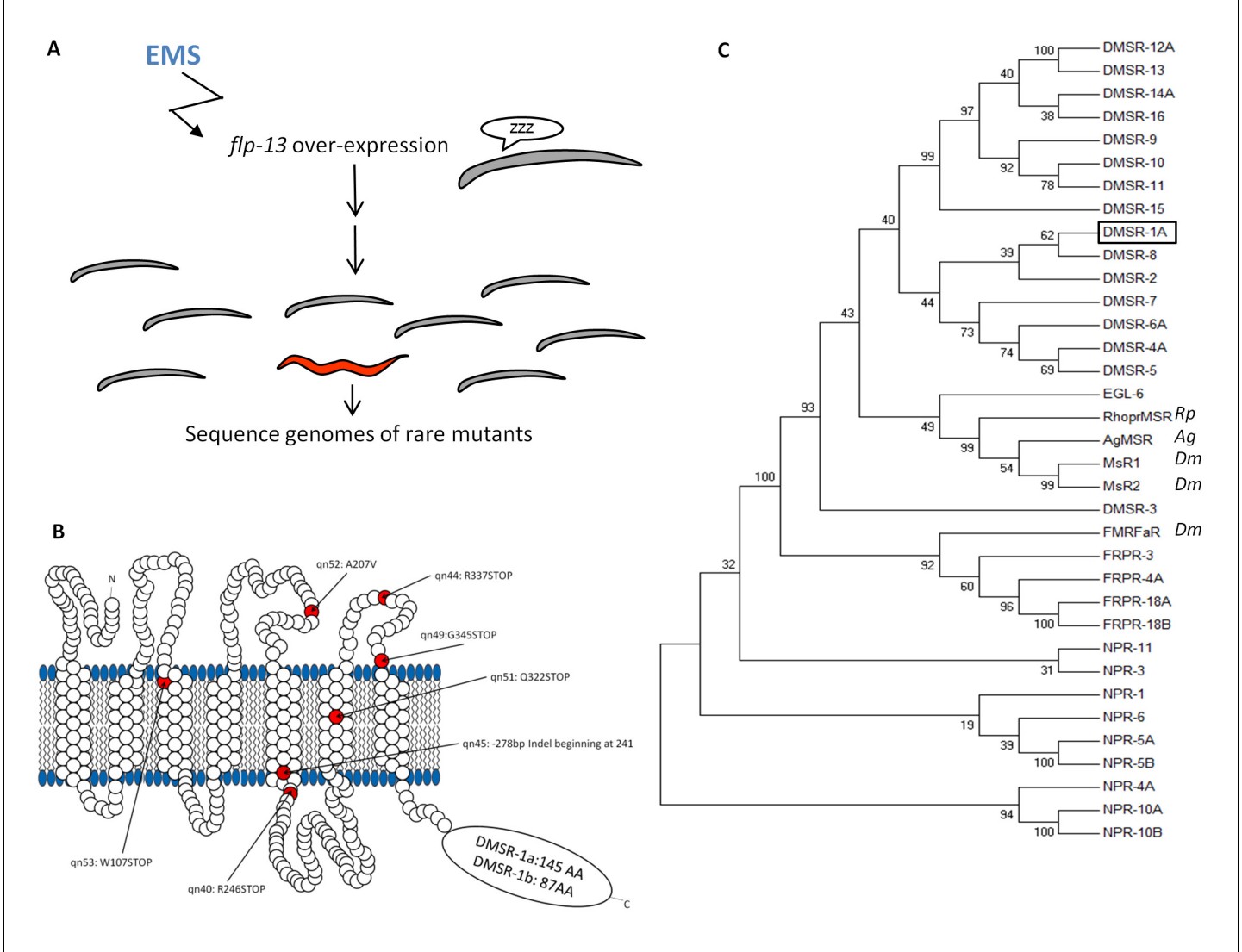

**Figure 1.** Mutations in the seven-transmembrane domain protein DMSR-1 suppress *flp-13* induced quiescence. (**A**) Mutagenesis approach to identify the downstream signaling mechanism for FLP-13 peptides. The grand-daughters of *Phsp-16.2:flp-13* worms that were mutagenized with the chemical ethyl methanesulfonate (EMS) were screened for rare animals that continued to feed and move after induction *flp-13* overexpression. Moving animals were selected using a microfluidics automated assay (*Yuan et al., 2015*) and feeding animals were identified by direct observation of pharyngeal pumping movements. (**B**) The DMSR-1 protein is predicted to have seven transmembrane domains with a C-terminus tail of either 145 amino acids (isoform A) or 87 amino acids (isoform B). Five mutations in DMSR-1 result in premature stop codons, one mutation results in an alanine to valine change in the second extracellular loop, and one mutation results in the removal of the C-terminal half of the protein. (**C**) Phylogenetic tree relationship between 30 *C. elegans* proteins previously demonstrated or predicted to be RFamide receptors, three *Drosophila melanogaster* RFamide receptors (NM_139501; NP_647713; NP_647711), and one receptor from each *Anopheles gambiae* (XP_314133) and *Rhodnius prolixus* (*Lee et al., 2015*). Unless indicated otherwise, all proteins are from *Caenorhabditis elegans*. We drew a box to highlight DMSR-1 (isoform A). The evolutionary history was inferred by using the Maximum Likelihood method based on the JTT matrix-based model (*Jones et al., 1992*). The bootstrap consensus tree inferred from 1000 replicates is taken to represent the evolutionary history of the taxa analyzed. The percentage of replicate trees in which the associated taxa clustered together in the bootstrap test (1000 replicates) is shown next to the branches. Initial tree(s) for the heuristic search were obtained automatically by applying Neighbor-Join and BioNJ algorithms to a matrix of pairwise distances estimated using a JTT model, and then selecting the topology with superior log likelihood value. The analysis involved 35 amino acid sequences. All positions containing gaps and missing data were eliminated. There were a total of 181 positions in the final dataset. Evolutionary analyses were conducted in MEGA7 (*Kumar et al., 2016*).

The following figure supplement is available for figure 1:

**Figure supplement 1.** DNA rearrangement of *dmsr-1* exon 4 in the *qn45* allele.

receptors from the arthropods *Anopheles gambiae* and *Rhodnius prolixus,* which also share homology to the Drosophila myosuppressin receptors. Based on its mutant phenotype and its homology to known RFamide receptors, we hypothesized that DMSR-1 is a receptor for FLP-13 neuropeptides.

To measure the degree to which these *dmsr-1* mutations suppressed the behavioral quiescence induced by *flp-13* overexpression, we subjected *Phsp-16.2:flp-13* transgenic animals to a heat pulse. We quantified feeding and locomotion quiescence two hours after induction of *flp-13* overexpression – well after the acute effects of heat on quiescence have dissipated (*Nelson et al., 2014*). All wild-type animals pumped their pharynxes rapidly two hours after the heat pulse whereas the *Phsp-16.2:flp-13* transgenic animals showed little to no feeding movements (*Figure 2A–B*). By contrast, a significant fraction of each of the *dmsr-1* mutants had pharyngeal pumping movements despite overexpression of *flp-13* (*Figure 2A*). In the absence of a heat pulse, *dmsr-1(qn45)* mutants pumped rapidly at a rate no different from wild-type animals (*Figure 2B*). Following induction of *flp-13* overexpression, the rate of pumping in *dmsr-1(qn45)* mutants was not changed.

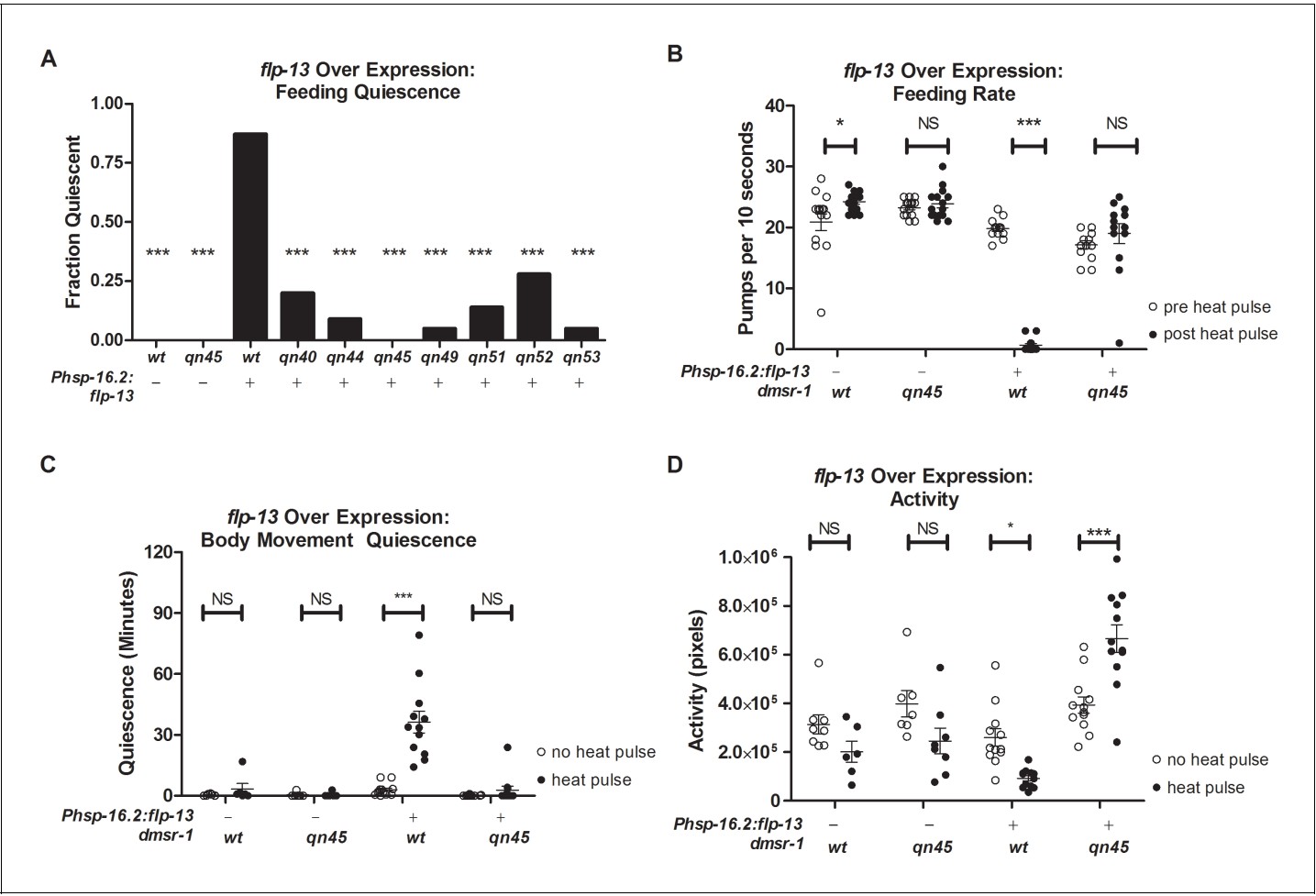

**Figure 2.** *dmsr-1* mutations suppress *flp-13* induced quiescence. (**A**) Fraction of animals quiescent for pharyngeal pumping two hours after induction of *flp-13* overexpression by exposure to a 30 min 33°C heat pulse. Statistical significance was assessed using a Fisher's exact test. N = 15–20 for each genotype. Asterisks indicate significant difference (p<0.0001) compared to wild type animals two hours after heat pulse. (**B**) Rate of pharyngeal pumping in wild type or *dmsr-1(qn45)* mutants with or without the *Phsp16.2:flp-13* transgene comparing the behavior pre-heat pulse to two hours post-heat pulse. (**C**) Body movement quiescence in a two hour period starting two hours after induction of *flp-13* overexpression. (**D**) Total body movement activity in the same two-hour period as (**C**). Activity is the sum of pixels changed between sequential images. Statistical significance in panels B-D was assessed using a 2-Way ANOVA with post-hoc pairwise comparisons made using Bonferroni correction method. Error bars denote Mean ± SEM. *p<0.05, **p<0.01, ***p<0.001.

To quantify body movements, we recorded worm activity and quiescence using the WorMotel (*Churgin and Fang-Yen, 2015*). The WorMotel is a polydimethylsulfoxane (PDMS) device consisting of 48 individual wells filled with agar nematode growth medium (NGM) and used to track individual worms over several hours. Two hours after induction of *flp-13* overexpression, we recorded body movement activity for two hours by taking images every 10 s. We used image subtraction analysis (*Raizen et al., 2008*) to determine both the total activity of the animals and the amount of time that each worm spent quiescent following induction of *flp-13* overexpression. Following induction of *flp-13* overexpression, *dmsr-1(qn45)* mutants had increased body movement activity and reduced body movement quiescence in comparison to animals that were wild type for *dmsr-1* (*Figure 2C–D*).

We can make a number of predictions based on the hypothesis that DMSR-1 is the receptor for FLP-13 neuropeptides. The first prediction is that, like *flp-13* (*Nelson et al., 2014*), *dmsr-1* should be required for quiescence during SIS. The second prediction is that FLP-13 neuropeptides will activate DMSR-1 in a cell-culture system. The third prediction is that *dmsr-1* should be expressed in the nervous system. Finally, DMSR-1 should be active in neurons that affect sleep/wake behavior.

## DMSR-1 is required for stress induced sleep

In order to test whether *dmsr-1* is required for SIS, we first artificially activated the SIS pathway. In response to stress, the *C. elegans* EGF (called LIN-3) activates the EGF-receptor (called LET-23) on the ALA neuron (*Van Buskirk and Sternberg, 2007*; *Hill et al., 2014*). Overexpression of LIN-3/EGF produces a robust sleep state that mimics SIS (*Van Buskirk and Sternberg, 2007*). While *dmsr-1* mutants showed wild-type levels of feeding quiescence in response to EGF overexpression (*Figure 3A*), they had a small but highly significant defect relative to wild-type worms in body movement quiescence (*Figure 3B*). This result provides evidence that *dmsr-1* is required downstream of or in parallel to EGF. The observation of only partial suppression of the EGF-induced quiescence suggests that there are other ALA-regulated signaling pathways working in parallel to *dmsr-1*.

To test the hypothesis that *dmsr-1* mediates SIS, we induced sleep by exposing the animals to a heat stress of 35°C for 30 min. *dmsr-1* mutations suppressed SIS with respect to both body movement quiescence and feeding quiescence (*Figure 3C–D*). To test whether *dmsr-1* was required for SIS after exposure to other temperatures, we exposed cohorts of animals to temperatures ranging from 29 to 37 degrees Celsius. The *dmsr-1(qn45)* mutation suppressed SIS at temperatures ranging from 34 to 37 degrees (*Figure 3—figure supplement 1A*). To test whether *dmsr-1* was required for sleep in response to stressors other than heat, we exposed animals to ultraviolet C (UVC) radiation. UVC induced a strong body movement quiescent response, which was attenuated in the *dmsr-1 (qn45)* mutant (*Figure 3—figure supplement 1B*). We partially restored SIS in *dmsr-1* mutants by expressing a genomic fragment of *dmsr-1* including the coding region and 4 kb of upstream regulatory DNA (*Figure 3C,D*). It is possible that the incomplete rescue of SIS is due to an absence of *dmsr-1* in the full complement of cells where it acts. This incomplete expression may be due to the nature of our extrachromosomal DNA array, which is not inherited by every somatic cell. Alternatively, it is possible that the genomic DNA fragment does not contain all required *dmsr-1* regulatory elements. While *dmsr-1* mutants were defective in quiescence associated with SIS, they had normal quiescence associated with DTS (*Figure 3—figure supplement 1C–D*), as previously shown for *flp-13* mutants (*Nelson et al., 2014*), supporting the notion that the DTS and SIS are regulated at least partially differently (*Trojanowski et al., 2015*).

These experiments therefore demonstrate that, in response to cellular stress, *dmsr-1* is required for quiescence of feeding movements and body movements.

## FLP-13 neuropeptides activate DMSR-1 in cell culture cells

To test the prediction that FLP-13 peptides activate DMSR-1 in cell culture, we cloned DMSR-1 isoform A into a mammalian expression vector and transiently expressed the protein in Chinese hamster ovarian (CHO) culture cells. These cells stably express a promiscuous Gα16 subunit as well as an aequorin reporter activated by intracellular calcium fluxes. Gα16 causes a $Ca^{2+}$ flux in response to receptor activation regardless of the type of G-protein that couples to the receptor in vivo (*Figure 4A*) (*Beets et al., 2011*).

We measured the response of cells expressing DMSR-1 isoform A to a range of FLP-13 peptide concentrations. Since the FLP-13 protein is processed into seven distinct RFamide peptides

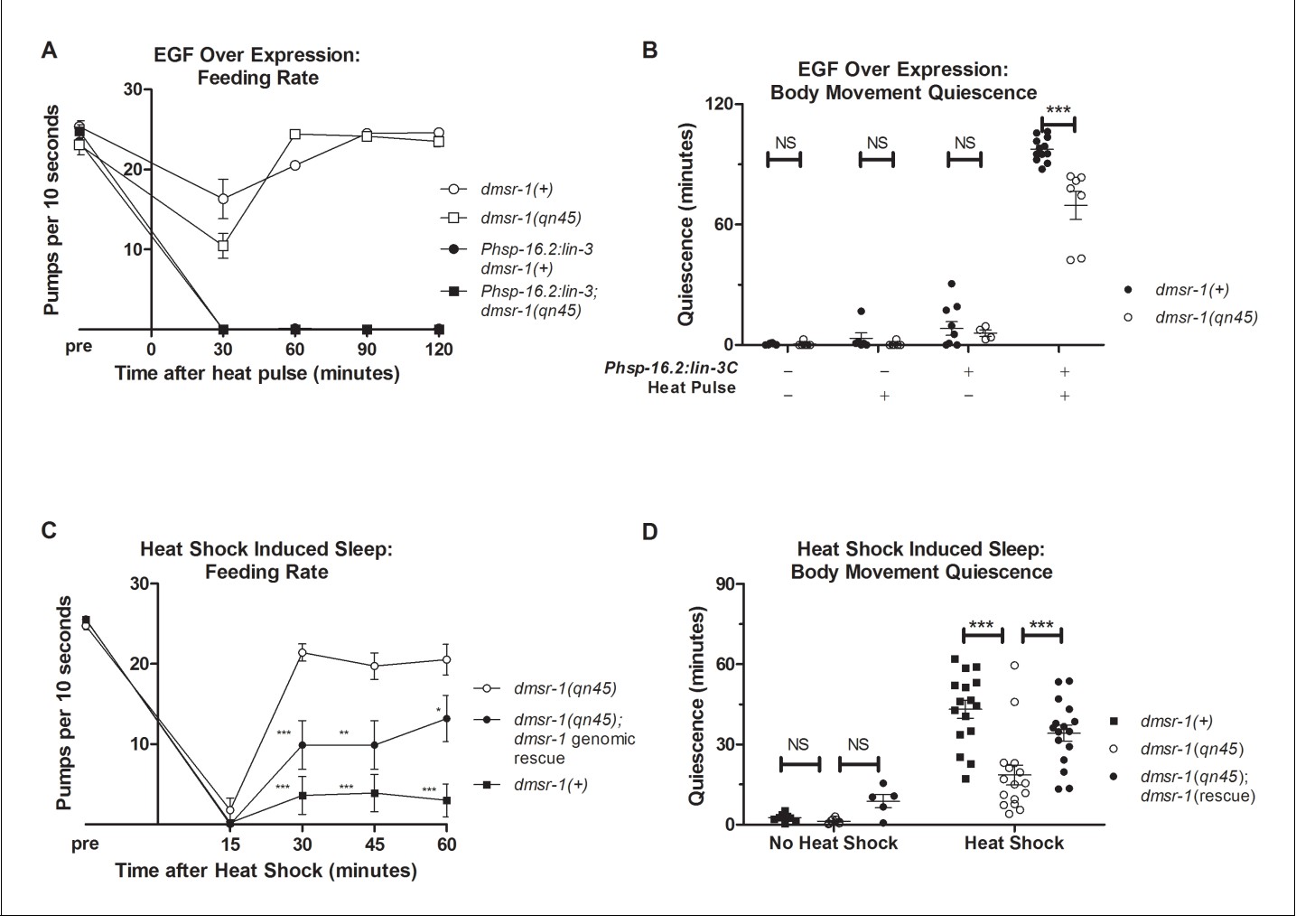

**Figure 3.** *dmsr-1* mutants are defective in quiescence associated with stress-induced sleep. (**A**) Rate of pharyngeal pumping before and up to two hours following heat pulse induction of EGF/LIN-3C overexpression. *dmsr-1* does not suppress EGF induced feeding quiescence. (**B**) *dmsr-1(qn45)* partially suppresses body movement quiescence induced by EGF overexpression (*Phsp-16.2:lin-3*). Body movements were measured for two hours starting one hour after induction of EGF overexpression. (**C**) Rate of pharyngeal pumping in *dmsr-1(qn45)* mutants and *dmsr-1* genomic rescue during the first hour following 35°C heat shock to induce SIS. Rescue construct is the operon-based reporter shown in *Figure 5*. Asterisks denote significant difference compared to *dmsr-1(qn45)*. (**D**) Body movement quiescence during 90 min after a 35°C heat shock. The *dmsr-1(qn45)* mutation suppresses body movement quiescence in response to heat shock. This defect in quiescence is rescued by a genomic fragment containing *dmsr-1*. Statistical significance was assessed using a 2-Way ANOVA with post-hoc pairwise comparisons made using Bonferroni correction method. Error bars denote Mean ± SEM. *p<0.05, **p<0.01, ***p<0.001.

The following figure supplement is available for figure 3:

**Figure supplement 1.** *dmsr-1* mutants are defective in SIS triggered by different stressors but are not defective in developmentally timed sleep during lethargus.

(*Figure 4B*), we performed a dose-response study with each of these seven peptides. Each of these peptides was capable of activating DMSR-1. The most potent activator was FLP-13–5, which elicited a 50% maximal response (EC$_{50}$) at a concentration of 2.3 nM (*Figure 4B–C*). The observed potent activation of DMSR-1 by FLP-13 peptides in this cell based assay supports the hypothesis that DMSR-1 is an *in vivo* receptor for FLP-13 peptides.

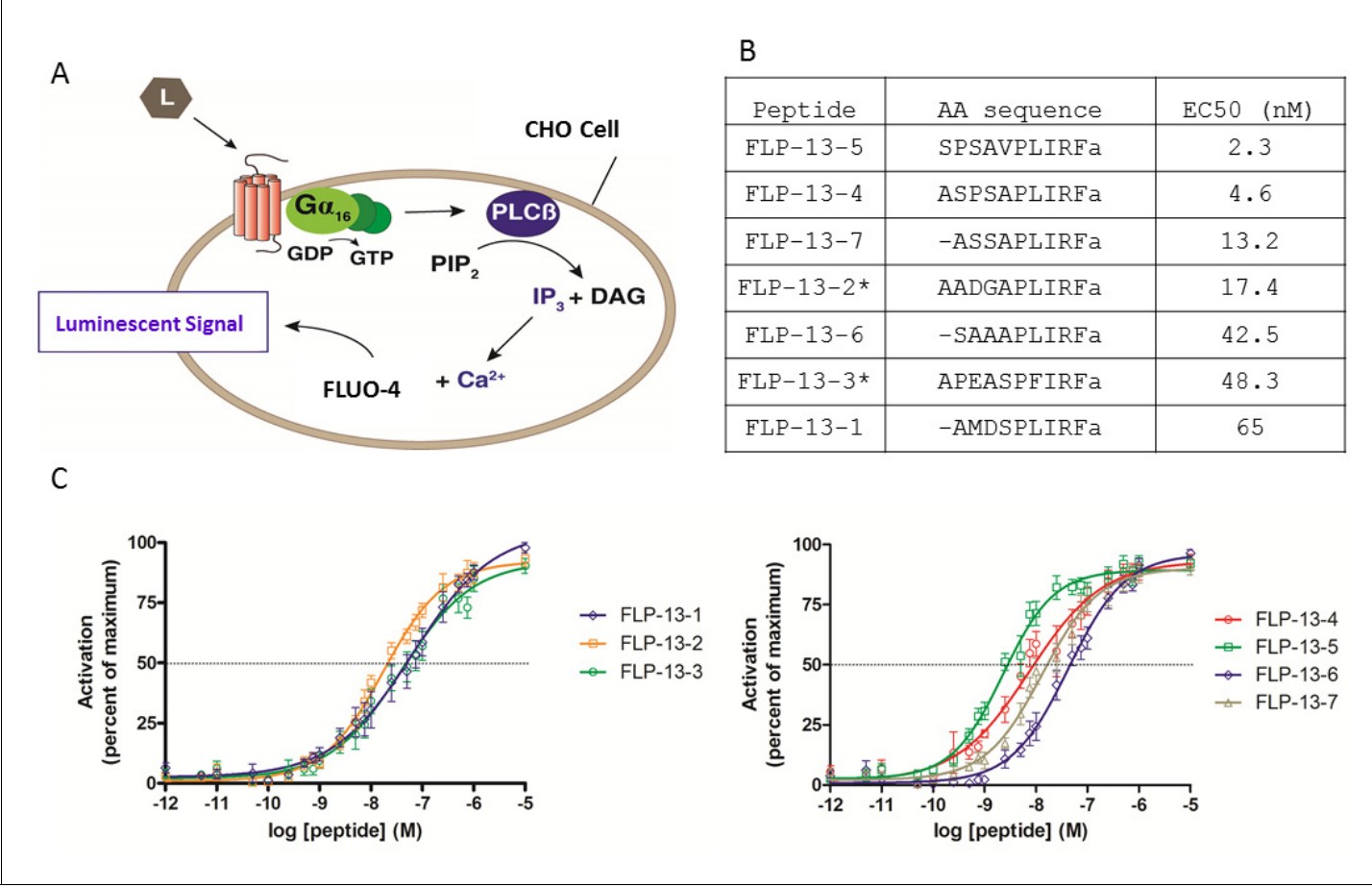

**Figure 4.** FLP-13 peptides activate DMSR-1 in a cell-based system. (**A**) Signal transduction components in a cell system used to test for receptor activation by peptide ligands (L). DMSR-1A was expressed in CHO cells along with the calcium sensitive bioluminescent protein aequorin. The receptor was paired with the promiscuous human $G_{\alpha16}$ protein, which causes calcium release from intracellular storage sites upon receptor activation. The calcium response elicits a luminescent signal from aequorin, a calcium-activated protein. (**B**) FLP-13 peptide sequences. *flp-13* encodes seven distinct peptides (* indicates that this peptide is encoded by the gene sequence twice). $EC_{50}$ indicates the concentration of neuropeptide required to elicit 50% of the maximum luminescent response from the aequorin protein (as shown in **C**). (**C**) Dose response curves for the seven FLP-13 peptides. Error bars represent SEM from 8–12 trials of neuropeptide treatment. Line represents non-linear regression fit of a variable slope line using four parameters. The x-axis is shown on a logarithmic scale.

## DMSR-1 is expressed in the nervous system

The third prediction made by our model is that *dmsr-1* is expressed in the nervous system. This prediction arises from prior studies indicating that the mechanism of feeding and movement quiescence observed with *flp-13* overexpression occurs through the inhibition of cholinergic motor neurons (*Trojanowski et al., 2015*). To test this prediction, we attached the rescuing genomic DNA of *dmsr-1* to a trans-splice acceptor site followed by the coding sequence of the red fluorescent protein dsRED (*Figure 5A*). This *Pdmsr-1:dmsr-1:SL2:dsRED* transgene rescued the *dmsr-1* mutant phenotype (*Figure 3C–D*), indicating that the construct contains regulatory elements required for appropriate *dmsr-1* expression. We observed red fluorescence in several neurons both in the head and tail of the animals (*Figure 5B*). Based on the location of the cell body, the neuronal process morphology, the co-expression of the transgene with other, well-characterized, green fluorescent protein (GFP) reporters, and the co-localization of the red fluorescence with green DiO fluorescence, we concluded that *dmsr-1* is expressed in the RID neuron (*Figure 5C*), the paired AIY neurons (*Figure 5D*), 12 other head neurons, the paired PHA and PHB neurons in the tail (*Figure 5E*), and in five other tail neurons.

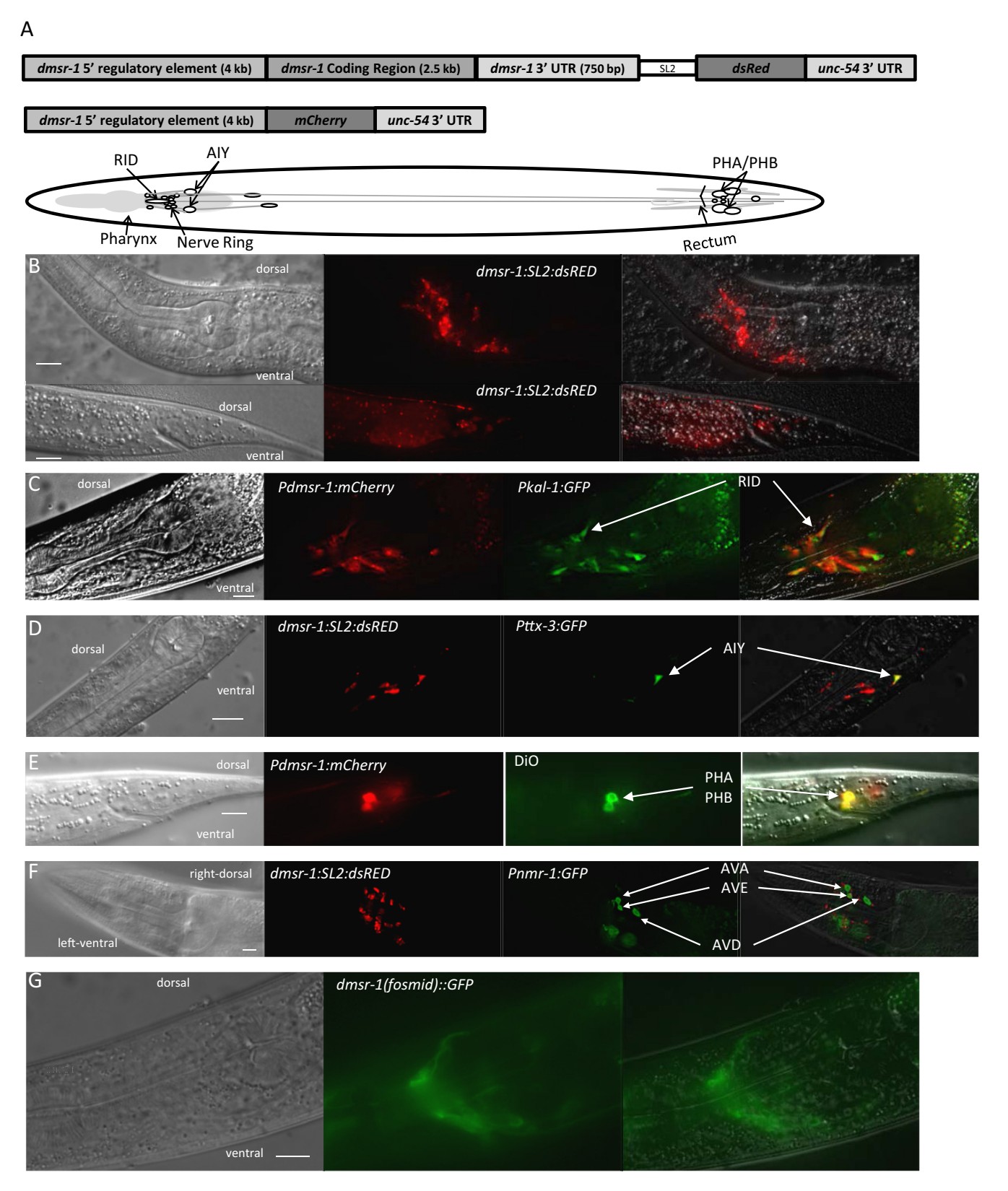

**Figure 5.** DMSR-1 is expressed non-synaptically in the nervous system. (**A**) Schematic of operon (top) and promoter fusion (middle) transcriptional reporters used to determine expression pattern of *dmsr-1*. Schematic overview of *dmsr-1* expression pattern, with identified neurons labelled. (**B**)

*Figure 5 continued on next page*

*Figure 5 continued*

Example images of *dmsr-1* expression in head and tail. The images were processed by 3D deconvolution (Leica Application Suite X, Leica), and presented as a maximum projection of a Z-stack. In this and subsequent images, anterior is to the left. (**C**) Colocalization of *dmsr-1* promoter reporter with *kal-1* (**Wenick and Hobert, 2004**) in the RID neuron. The *dmsr-1* promoter:mCherry reporter gave the same expression pattern as the operon-based transcriptional reporter shown in panel A. (**D**) Colocalization of *dmsr-1* mCherry reporter with *ttx-3* (**Hobert et al., 1997**) in an AIY neuron. Images were captured as a z-stack and processed by 3D deconvolution. One individual slice is shown. (**E**) DiO staining of tail phasmid neurons PHA and PHB, which colocalizes with the *dmsr-1* red reporter. (**F**) Lack of colocalization of *dmsr-1* promoter reporter with *nmr-1* (**Brockie et al., 2001**) in command interneurons. Images were captured as a z-stack and processed by 3D deconvolution. One deconvolved slice, which shows all three neuron types, is shown. There is a red neuron that partially overlaps the green AVE neuron but our close evaluation of multiple worms shows that these neurons are distinct. (**G**) Membrane localization of GFP attached to the C-terminus of DMSR-1 using a fosmid construct (TransgeneOme project).

The following figure supplements are available for figure 5:

**Figure supplement 1.** Rescue of the *dmsr-1* mutant quiescence-defective phenotype with a fosmid that encodes a GFP tag at the c-terminus of DMSR-1.

**Figure supplement 2.** Effects of selective expression of *dmsr-1* in the paired AIY head neurons or in the paired PHA tail neurons.

Based on additional analysis, we concluded that *dmsr-1* is not expressed in several other neurons, which are listed in *Supplementary file 1*. Importantly, we did not see expression of *dmsr-1* in AVE or AVA neurons (*Figure 5F*). These command interneurons are the only post-synaptic partners to the ALA neuron based on the presence of ultrastructurally-defined chemical synapses (*White et al., 1986*). The expression in neurons that do not make direct synaptic connections with ALA suggested that ALA signals in a neuroendocrine fashion to promote SIS. In support of this suggestion, we found that a DMSR-1 protein with a GFP fluorescent tag at its C-terminus was localized diffusely to neuronal cell membranes and was not localized to synapses (*Figure 5G*). This GFP-tagged DMSR-1 protein rescued the mutant phenotype of *dmsr-1* (*Figure 5—figure supplement 1*), indicating that it was functional.

Taken together, these results support a model whereby ALA releases FLP-13 peptides in a neuroendocrine fashion to directly activate DMSR-1 expressed in neurons and thus promote sleep in response to cellular stress. The notion that ALA can signal via non-synaptic means was previously proposed based on the observation that a mutant with disrupted ALA axonal outgrowth retained normal ALA function (*Van Buskirk and Sternberg, 2007*).

In an attempt to determine which of the neurons observed to express *dmsr-1* is sufficient to relay the sleep inducing effect of *flp-13* overexpression, we transgenically restored *dmsr-1* gene expression in specific neurons in animals otherwise lacking *dmsr-1* function. We tested two neuron types—AIY and PHA—where we had the tools to easily express DMSR-1 selectively in a single neuron type. Expression of *dmsr-1* in the paired PHA tail sensory neurons was not sufficient to restore feeding or body movement quiescence induced by *flp-13* overexpression (*Figure 5—figure supplement 2A–C*). Expression of *dmsr-1* in the paired head AIY interneurons conferred a small but statistically-significant (p<0.01) rescue of the body movement quiescence defect (*Figure 5—figure supplement 2D*) but not the feeding quiescence defect observed in *dmsr-1* mutants during *flp-13* overexpression (*Figure 5—figure supplement 2A–B*). AIY specific *dmsr-1* expression did not however rescue the feeding or body movement quiescence defects observed in *dmsr-1* mutants during SIS (*Figure 5—figure supplement 2E–F*).

## DMSR-1 acts to inhibit neuronal activity

The fourth prediction from our model is that DMSR-1 signaling affects neurons that control sleep/wake behavior. DMSR-1 may regulate sleep either by activating sleep-promoting neurons or by inhibiting wake-promoting neurons. To distinguish between these possibilities, we inhibited the *dmsr-1*-expressing cells to test their effects on sleep. If DMSR-1 activation promotes sleep by activating sleep-promoting neurons, then inhibiting these neurons should impair sleep. Conversely, if DMSR-1 activation promotes sleep by inhibiting wake-promoting neurons, then inhibiting these cells would promote sleep.

We inhibited DMSR-1 expressing neurons by using histamine-gated chloride (HisCl) channels. We can use this approach because histamine does not appear to act as a neurotransmitter in *C. elegans* (*Hobert, 2013*). In the presence of histamine, HisCl channels allow chloride ions to pass into the neurons to render them less excitable (*Liu and Wilson, 2013*; *Nelson et al., 2014*; *Pokala et al., 2014*). By applying histamine to worms expressing HisCl channels under the control of the *dmsr-1* promoter (*Pdmsr-1:HisCl*), we inhibited *dmsr-1* expressing neurons in a spatially- and temporally- controlled fashion. Because non-transgenic animals lack endogenous receptors for histamine (*Pokala et al., 2014*), they behaved similarly in the presence and absence of histamine (*Figure 6*). When transgenic worms expressing *Pdmsr-1:HisCl* worms were placed on histamine, their movement quiescence and total activity were not affected (*Figure 6*), indicating that in the absence of cellular stress, silencing these neurons has minimal effect on behavior.

We then tested whether activating the HisCl channels affected activity and quiescence after triggering cellular stress. For these experiments, we triggered cellular stress by exposing the animals to ultraviolet (UV) light (*Figure 3—figure supplement 1*). UV light exposure triggers a robust sleep state that requires ALA and FLP-13 signaling (H. Debardeleben and D. Raizen, unpublished data). The long-lasting effects of UV induced sleep provide a sensitive tool for measuring the effects of neuronal silencing in this experiment. When HisCl worms were placed on histamine following exposure to UV stress, they had reduced total activity and enhanced body movement quiescence compared to control worms that lacked the HisCl channels (*Figure 6A–C*). Importantly, HisCl channels did not affect activity or quiescence in the absence of histamine.

These observations of elevated body movement quiescence with HisCl activation suggest that DMSR-1 promotes sleep by inhibiting wake-promoting cells.

### *dmsr-1* and *flp-13* have parallel activity

One neuropeptide (or a group of neuropeptides encoded by the same gene) may activate multiple receptors. Similarly, a given receptor may be activated by multiple neuropeptides. Does the somnogenic effect of FLP-13 neuropeptides result exclusively from the activation of DMSR-1? Do other neuropeptides activate DMSR-1 to induce sleep? We used a double mutant analysis to answer these questions. We compared the SIS phenotypes of *flp-13; dmsr-1* double mutant animals to those of *dmsr-1* and *flp-13* single mutant animals. We reasoned that if *flp-13* and *dmsr-1* have sleep inducing effects that are independent of one another, then the double mutant will have a greater defect in SIS compared to either of the single mutants. The double mutant animals indeed showed a greater defect in quiescence than either the *dmsr-1* or the *flp-13* single mutant animals (*Figure 7*). This result suggests that (1) FLP-13 peptides activate receptors in addition to DMSR-1, and (2) DMSR-1 is either activated by ligands in addition to those encoded by *flp-13*, or DMSR-1 has ligand-independent activity.

### Discussion

Our findings expand our understanding of how stress induced sleep is regulated in *C. elegans* (*Figure 8*). Cellular stress leads to the release of the EGF/LIN-3 (*Hill et al., 2014*) either from the stressed cells directly or from other cells. EGF activates the ALA neuron through the EGF-receptor LET-23 (*Van Buskirk and Sternberg, 2007*). This activation requires depolarization of ALA (*Nelson et al., 2014*). ALA releases neuropeptides encoded by the *flp-13* gene (*Nelson et al., 2014*) as well as other neuropeptides (*Nath et al., 2016*). FLP-13 neuropeptides then act in a neuroendocrine fashion to activate the G-protein coupled receptor DMSR-1 and thus to inhibit wake-promoting neurons, which include the AIY interneurons. The G-protein that is coupled to DMSR-1 may be the Gi/o alpha subunit GOA-1, since *goa-1* mutants are defective in SIS and in *flp-13* induced sleep (*Trojanowski et al., 2015*) and since *goa-1* is expressed widely in the nervous system (*Ségalat et al., 1995*). GOA-1 signaling may ultimately down regulate the activity of the Gq (EGL-30) signaling pathway, since a gain of function mutation in EGL-30 suppresses *flp-13* induced sleep (*Trojanowski et al., 2015*).

The observation that the suppression of quiescence by most of our genetic manipulations is only partial suggests that there are additional complexities in the sleep promoting system downstream of ALA activation. Our double mutant analyses support the notion that FLP-13 peptides act on other receptors in addition to DMSR-1, and that DMSR-1 is activated by peptides other than FLP-13

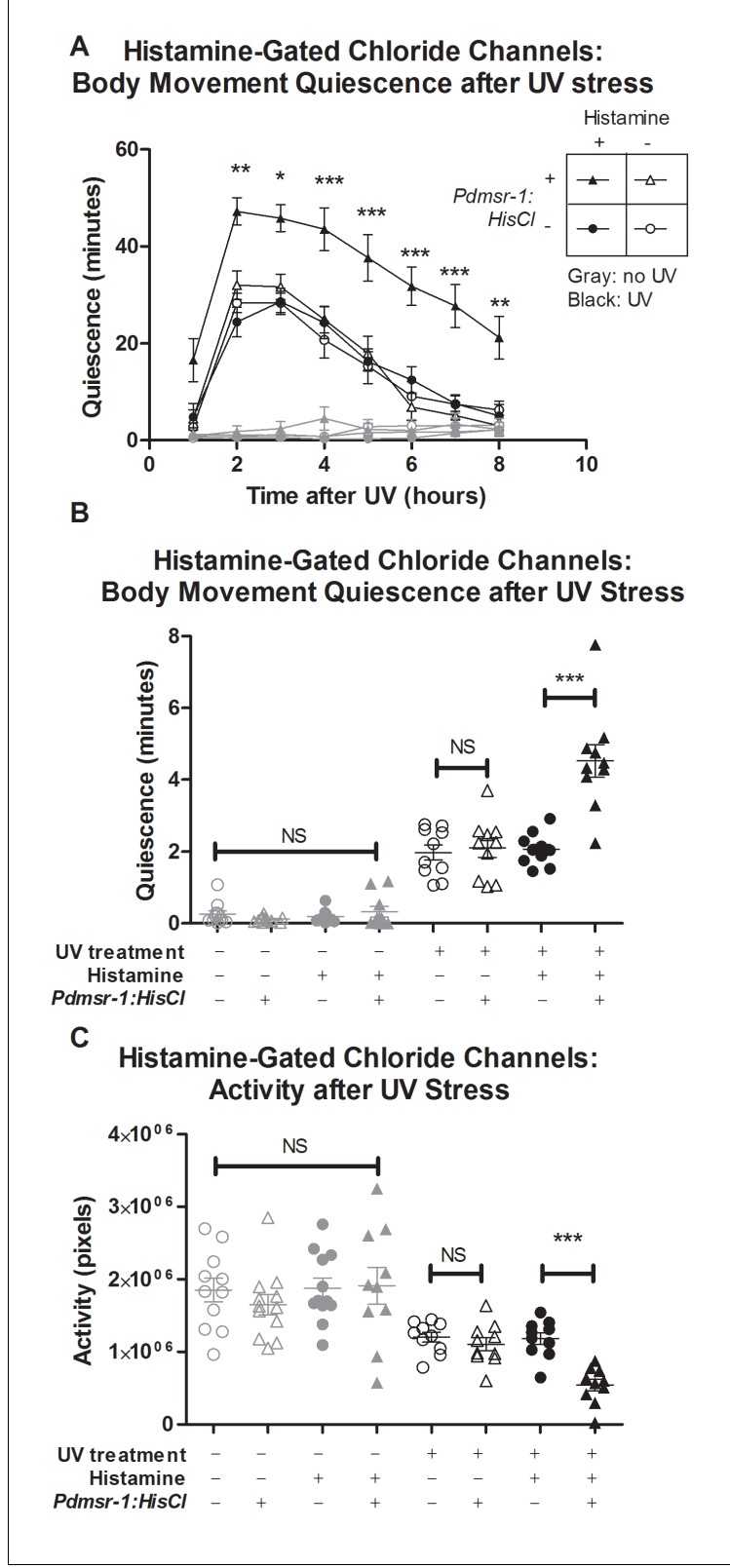

**Figure 6.** DMSR-1 has an inhibitory effect on neurons. (**A**) *Pdmsr-1:HisCl* worms were placed on histamine-containing agar, and sleep was induced by ultraviolet C (UVC) light irradiation. Worms were compared to the wild type strain (N2) on histamine, as well as to worms not exposed to histamine. Gray lines indicate control worms that were not exposed to UVC light, whose quiescence was not affected by activation of HisCl channels. Statistical

*Figure 6 continued on next page*

*Figure 6 continued*

comparisons were performed using a 2-way repeated measures ANOVA using time and experimental group as factors. Asterisks indicate lowest level of significance in comparisons between *Pdmsr-1:HisCl* on histamine with each of the other groups in pairwise comparison using Bonferroni correction. *p<0.05, **p<0.005, ***p<0.0005. (B) Total amount of sleep over eight hours from *Figure 6A* for each individual worm. There was a significant interaction between histamine and HisCl channels such that HisCl channels only increased sleep in the presence of histamine. ***p<0.0005. There were no significant effects found for worms that were not irradiated with UV. NS denotes not significant. (C) Total activity analysis across an eight-hour period. HisCl channels reduced activity only in animals exposed to histamine, and there were no significant effects for worms that were not irradiated with UV. ***p<0.0005. NS denotes not significant. In all three panels, triangles represent animals that expressed Pdmsr-1: HisCl, circles represent animals that do not express Pdmsr-1:HisCl, filled symbols represent animals that have been exposed to histamine, empty symbols represent animals that have not been exposed to histamine, black symbols represent animals that have been exposed to UV irradiation, and gray symbols represent animals that were not exposed to UV irradiation. In panels B and C, statistical significance was assessed with a 2×2 Factorial ANOVA with Bonferroni post-hoc correction.

peptides. The additional receptors responding to FLP-13 peptides may include one or more of the 15 other DMSR proteins encoded in the *C. elegans* genome (*Figure 1C*), and the additional peptides acting on DMSR-1 may include one or more of the other neuropeptides released from ALA (*Nath et al., 2016*).

We showed that selective *dmsr-1* expression in the AIY neurons conferred a small but significant rescue of the *flp-13* overexpression-induced movement quiescence but did not confer a rescue of SIS. Three possibilities could reconcile these observations. (1) The endogenous FLP-13 peptides that act on the AIY neurons are released from a neuron other than ALA. (2) FLP-13 peptides act on AIY only when expressed at high levels and do not normally do so under physiological conditions. (3) Our SIS assay conditions were not sufficiently sensitive to identify a small rescue in SIS. Regardless of the explanation of these results, since the AIY interneurons are known to promote worm locomotion (*Gray et al., 2005*; *Shtonda and Avery, 2006*), our observations that *dsmr-1* is expressed in AIY

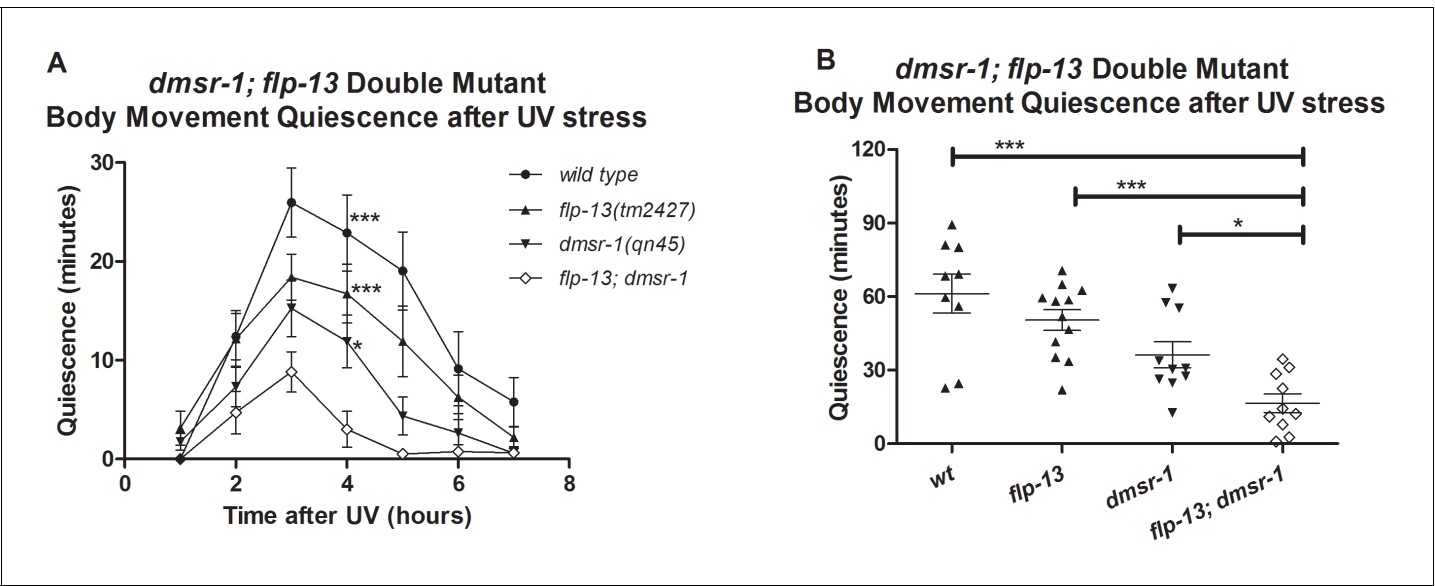

**Figure 7.** DMSR-1 and FLP-13 neuropeptides have parallel activity. (A) Double mutant analysis between *flp-13* and *dmsr-1*. Total amount of body movement quiescence was measured following UV irradiation. Statistical comparisons were made using 2-way repeated measures ANOVA with post-hoc pairwise Bonferroni correction method and restricted to comparisons made with the double mutant. (B) Total quiescence during first four hour period following UV irradiation shown in *Figure 7A*. Statistical comparisons were made using 1-way ANOVA with Dunnett's post-hoc pairwise comparisons with the double mutant.

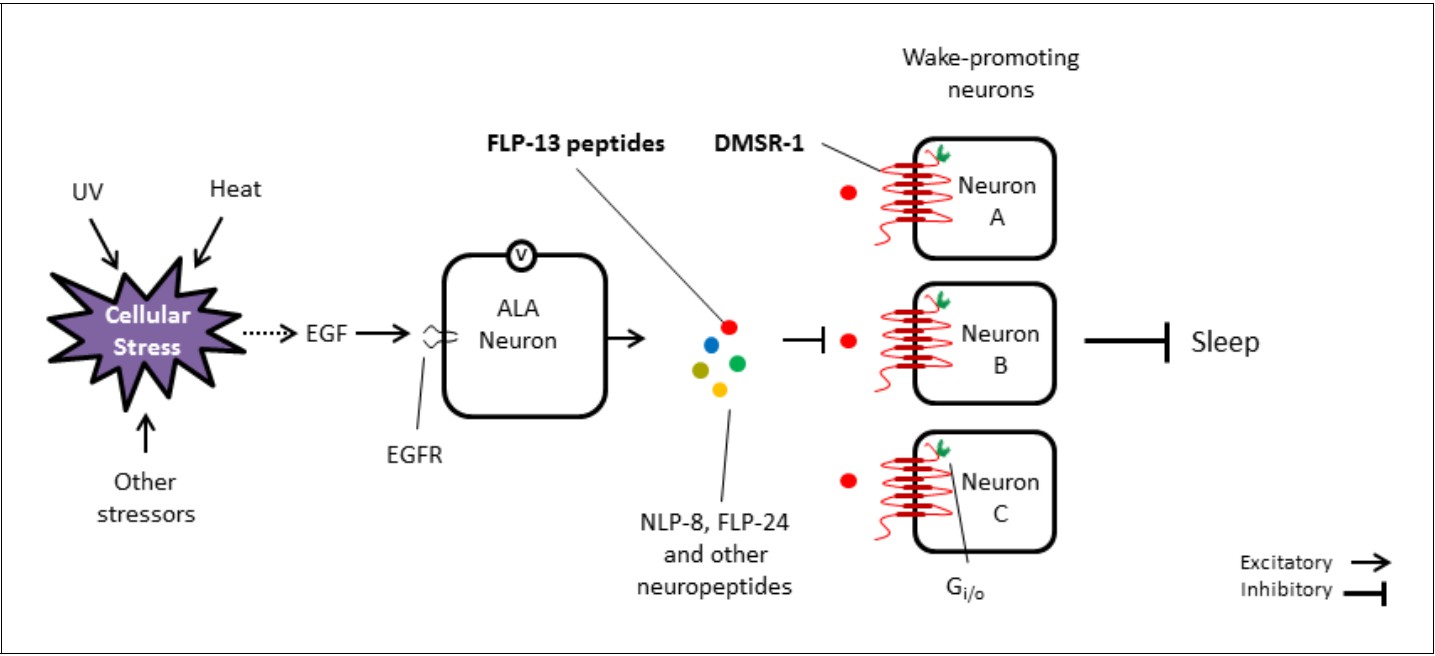

**Figure 8.** Model for the regulation of stress-induced sleep. Cellular stress leads to the release of EGF (LIN-3) either directly from the stressed cells or indirectly via other cells receiving a signal from the stressed cells. EGF activates the ALA neuron by binding to the EGF receptor (LET-23). ALA depolarizes and releases FLP-13 neuropeptides, among other sleep inducing signals. FLP-13 peptides signal in a non-synaptic fashion via the seven-transmembrane domain receptor DMSR-1 and the G protein alpha subunit Gi/o to inhibit several wake-promoting neurons. These neurons include AIY, PHA, PHB, RID, and other neurons.

and may function in AIY are consistent with the model that FLP-13/DMSR-1 signaling functions to inhibit wake-promoting neurons.

Although selective *dmsr-1* expression in neither AIY nor PHA neurons fully rescued the mutant quiescence phenotype, our findings do not exclude the possibility that *dmsr-1* functions selectively in a neuron type that we have not yet tested. Several of the other neurons expressing *dmsr-1* remain unidentified and may also promote waking activity. One possibility is the neuroendocrine RID neuron, which promotes forward locomotion (*Lim et al., 2016*). The alternative to the model that *dmsr-1* functions in a small subset of the neurons where it is expressed is that *dmsr-1* is simultaneously required in several neurons to promote sleep in response to FLP-13 signaling. In this model, which we favor, restoring *dmsr-1* gene expression in only a subset of the neurons in which *dmsr-1* is required would not be sufficient to fully rescue the mutant phenotype.

In mammals, wake-promoting centers are distributed broadly across brainstem, basal forebrain, midbrain, and diencephalic structures, but sleep promoting neurons are anatomically restricted (*Saper et al., 2005b*). The ventrolateral preoptic area of the hypothalamus, which is proposed to be sleep-promoting, contains inhibitory projections to several wake-promoting neurons (*Saper et al., 2005a*). It is possible that *C. elegans* similarly has a diffuse wake-promoting circuit that can be inhibited by a central sleep-promoting system, the single ALA neuron. It remains to be seen whether sleep in response to acute illness in other animals functions in a similar fashion, but since RFamide and EGF signaling regulates sleep in arthropods too, it would not be unreasonable to propose that the molecular details and signaling concepts elucidated here are conserved across phylogeny.

## Materials and methods

### Worm cultivation

Worms were cultivated on the agar surface of NGM medium (1.7% agar), fed the OP50 derivative bacterial strain DA837 (*Davis et al., 1995*), and maintained at 20 degrees Celsius unless noted

otherwise. We found that the type of agar used affected the behavior of animals after heat shock. For experiments reported here we used only granulated agar (Apex, catalog number 20–275).

## Strains used

N2

NQ570: *qnIs303[Phsp-16.2:flp-13; Phsp-16.2:GFP; Prab-3:Cherry]*

NQ793: *dmsr-1(qn40) V; qnIs303*

NQ810: *dmsr-1(qn44) V; qnIs303*

NQ792: *dmsr-1(qn45) V; qnIs303*

NQ814: *dmsr-1(qn49) V; qnIs303*

NQ51: *dmsr-1(qn51) V; qnIs303*

NQ52: *dmsr-1(qn52) V; qnIs303*

NQ53: *dmsr-1(qn53) V; qnIs303*

NQ915: *dmsr-1(qn45) V*

NQ602: *flp-13(tm2427)*

NQ943: *flp-13(tm2427) IV; dmsr-1(qn45) V*

PS5009: *pha-1(e2132ts) III (?); syEx723[Phsp16.2:LIN-3C; Pmyo-2:GFP; pha-1(+)]*

NQ978: *pha-1(e2132ts) III (?); dmsr-1(qn45) V; syEx723*

NQ990: *qnEx514[Pdmsr-1:HisCl; Pttx-3:GFP]*

NQ1006: *qnEx526[Pdmsr-1:mCherry; Punc-122:GFP; pha-1(+); ladder]*

NQ1083: *unc-119(ed3) III; qnEx585[dmsr-1(fosmid)::GFP; pCFJ151]*

NQ1055: *dmsr-1(qn45) V; qnEx569[dmsr-1(fosmid)::GFP; Pttx-3:GFP; 1 kb DNA ladder]*

NQ1111: *dmsr-1(qn45) V; qnEx602[dmsr-1:SL2:dsRed; Pttx-3:GFP; 1 kb DNA ladder]*

NQ1142: *dmsr-1(qn45); qnEx610[Pttx-3:dmsr-1:SL2:dsRed; Pmyo-2:GFP; 1 kb DNA ladder]*

NQ1145: *dmsr-1(qn45) V;qnIs303; qnEx610[Pttx-3:dmsr-1:SL2:dsRed; Pmyo-2:GFP; 1 kb DNA ladder];*

NQ1147: *dmsr-1(qn45) V; qnIs303; qnEx614[Pgcy-17:dmsr-1:SL2:dsRed;Pmyo-2:GFP; 1 kb DNA ladder]*

## Molecular biology

We used the overlap extension PCR method to generate constructs for transgenic analysis (*Nelson and Fitch, 2011*). Oligonucleotide sequences used for constructs and for sequencing are listed in *Supplementary file 2*.

Microinjection of extrachromosomal arrays was performed as described in *Mello et al. (1991)*.

Whole genome sequencing was performed at the Wistar Institute Genomics Core Facility and bioinformatics analysis of the sequences was performed at the Wistar Institute Bioinformatics Core Facility.

## Genetic crosses

Crosses with *dmsr-1* were performed using either the balancer nT1, which expresses a dominant GFP, or by following the *qn45* deletion by PCR analysis using the primers oNQ1480 and oNQ1515.

## Behavioral assays

Except for data shown in *Figure 3—figure supplement 1*, which was collected from fourth larval stage animals, all behavioral assays were performed in young adult worms. Worms were synchronized by picking larval stage four animals and performing the experiments the following day.

### SIS induction by heat

Sleep was induced by exposing the worms to 35°C for 30 min by immersion of their housing Petri dish sealed with Parafilm strips in a circulating water bath and then recovered at room temperature. For experiments in which we assessed feeding behavior by direct observation, we immersed the worms housed on the agar surface of 5.5 cm diameter polystyrene Petri dishes (Azer, Catalog numbers ES3515 and ES3514) containing 10 mL of NGM agar and seeded with DA837 bacteria. For experiments in which we measured body movements using the WorMotel, worms were

picked to the WorMotel chip and placed inside an empty plastic 10 cm diameter Petri dish, which was sealed with Parafilm and placed in the heat bath.

## Assessment of body movement activity and quiescence

The set-up for imaging the WorMotel is described in *Churgin and Fang-Yen (2015)*. The WorMotel is a PDMS device that contains a series of wells filled with NGM agar so that individual worms can be housed separately for longitudinal tracking. The chip was imaged under dark field microscopy using red light to indirectly illuminate worms from below a glass stage as described. Images were taken every ten seconds at a spatial resolution of 16 microns per pixel. Images were analyzed using custom software that subtracts pairs of successive images as previously described (*Raizen et al., 2008*). Body movement activity was quantified as the total number of pixels changed in the subtracted images, and quiescent epochs were those in which no pixel movement was detected between a pair of frames. The software is available at https://github.com/cfangyen/wormotel.

## SIS induced by ultraviolet light exposure

Sleep was induced by exposing worms to 1500 J/m$^2$ of UVC light (XL-UV Crosslinker, Spectrolinker Incorporated). Worms were picked to the WorMotel chip with a thin coating of bacteria for food and placed in a 10 cm petri dish with the lid removed inside the crosslinker. Behavior was assessed by recording on the WorMotel.

## Heat-induced overexpression

Overexpression of *flp-13* or of *lin-3* was induced in all somatic cells under the control of the heat shock inducible promoter *hsp-16.2*. Transcription was activated by immersing plates wrapped with Parafilm in a 33°C water bath for 30 min. To assess body movement quiescence, animals were exposed to heat while on NGM agar dishes, then transferred to the WorMotel chip and imaged. Feeding quiescence was measured two hours following induction of overexpression.

## Assessment of feeding behavior

Feeding behavior was assessed by examining pharyngeal pumping under a stereomicroscope using 2-4x objective magnification. Behavior was measured in real time, and the experimenter was blinded to the condition of the worm. A pump was identified as a backwards movement of the grinder, a tooth-like structure in the posterior bulb of the pharynx.

## Histamine-gated chloride channels

The coding region for the histamine-gated chloride channel was amplified using PCR from the pNP471 plasmid (*Pokala et al., 2014*), and combined with the *dmsr-1* promoter using overlap extension PCR. Worms were raised on NGM media and synchronized at the L4 stage. Young adults expressing *Pdmsr-1:HisCl* were placed in NGM agar medium with 15 mM histamine immediately prior to UV exposure and imaging on the WorMotel.

## Statistics

Multi-factor analyses were performed using 2-way ANOVA with pairwise post-hoc comparisons using the Bonferroni correction. For post-hoc tests, all comparisons were made against one group. For time course analyses, we used a 2-way ANOVA with the time factor as a repeated measure. One factor analyses were performed using a 1-way ANOVA with pairwise comparisons made using Dunnett's test. All statistics were performed and graphs were generated using GraphPad Prism 5.0.3.477 and 6.0 (GraphPad Software, Inc., La Jolla California).

## Receptor ligand interactions in cell culture

The assessment of FLP-13 peptide activation of DMSR-1 was done as described (*Nelson et al., 2015*). Briefly, DMSR-1A cDNA was cloned into the pcDNA3.1(+) TOPO expression vector (Thermo Fisher Scientific, Waltham Massachusetts). Receptor activation was studied in Chinese hamster ovary cells (CHO) stably expressing apo-aequorin (mtAEQ) targeted to the mitochondria as well as the human Galpha16 subunit. We used the CHO-K1 cell line (PerkinElmer, ES-000-A2) for receptor activation assays. Quality control and authentication was performed by determining the EC50 for

reference agonists (e.g. ATP) in an AequoScreen calcium mobilization assay. A mycoplasma test was performed using the MycoAlert Mycoplasma (Lonza) detection kit. This cell line tested negative for mycoplasma.

The CHO/mtAEQ/Galpha16 cells were cultured in Ham's F12 medium (Sigma), containing 10% fetal bovine serum (FBS), 100 UI/ml of penicillin/streptomycin, 250 µg/ml Zeocin and 2.5 µg/ ml Fungizone (Amphoterin B). Cell lines were grown at 37°C in a humidified atmosphere of 5% $CO_2$ and were diluted fifteen-fold every third day. CHO/mtAEQ/Galpha16 cells were transiently transfected with the DMSR-1 cDNA construct or the empty pcDNA3.1(+) vector using the Lipofectamine transfection reagent (Thermofisher Scientific), according to the manufacturer's instructions. Cells expressing the receptor were shifted to 28°C 1 day later, and collected 2 days post-transfection in BSA medium (DMEM/HAM's F12 with 15 mM HEPES, without phenol red, supplemented with 0.1% BSA) and loaded with 5 µM coelenterazine h (Thermo Fisher) for 4 hr to reconstitute the holo-enzyme aequorin. Cells (25,000 cells/well) were exposed to synthetic peptides in BSA medium, and aequorin bioluminescence was recorded for 30 s on a MicroBeta LumiJet luminometer (PerkinElmer, Waltham Massachusetts) in quadruplicate. For dose-response evaluations, after 30 s of ligand-stimulated calcium measurements, Triton X-100 (0.1%) was added to the well to obtain a measure of the maximum cell $Ca^{2+}$ response. BSA medium without the peptides was used as a negative control and 1 µM ATP was used to check the functional response of the cells. Cells transfected with the pcDNA3.1 empty vector were used as a negative control for the effect of the receptor. EC50 values were calculated from dose-response curves, constructed using a computerized nonlinear regression analysis, with a sigmoidal dose-response equation (Prism 6.0).

## Acknowledgements

We thank Jessie Zhou, Mark Nessel, and Eve Phelps for technical assistance, Hilary Debardeleben for showing that ultraviolet light causes SIS, and Gregg Artiushin for demonstrating feasibility of *flp-13(OE)* suppressor screen by isolating the first mutant. We thank Raizen lab members for discussions and comments on this manuscript. A HisCl plasmid was provided by Cori Bargmann. Some strains were provided by the CGC, which is funded by NIH Office of Research Infrastructure Programs (P40-OD010440). This research was supported in part by the National Institutes of Health (R01-NS088432, R21-NS091500, and P30-ES013508), the European Research Council (ERC-2013-ADG-340318), and the Research Foundation Flanders (FWO). Its contents are solely the responsibility of the authors and do not necessarily represent the official views of the NINDS, NIEHS, NIH, ERC, or the FWO.

## Additional information

### Funding

| Funder | Grant reference number | Author |
|---|---|---|
| National Institutes of Health | R01NS088432 | Michael J Iannacone<br>Christopher Fang-Yen<br>David M Raizen |
| National Institutes of Health | R21NS091500 | Michael J Iannacone<br>Christopher Fang-Yen<br>David M Raizen |
| European Research Council | ERC-2013-ADG-340318 | Isabel Beets<br>Liliane Schoofs |
| Fonds Wetenschappelijk Onderzoek | | Isabel Beets<br>Liliane Schoofs |
| National Institutes of Health | P30ES013508 | David M Raizen |
| NIH Office of Research Infrastructure | P40-OD010440 | David M Raizen |

The funders had no role in study design, data collection and interpretation, or the decision to submit the work for publication.

## Author contributions

MJI, Conceptualization, Data curation, Investigation, Methodology, Writing—original draft, Writing—review and editing; IB, LEL, Investigation, Writing—original draft, Writing—review and editing; MAC, CF-Y, Software; MDN, Investigation, Writing—review and editing; LS, Conceptualization, Writing—review and editing; DMR, Conceptualization, Data curation, Funding acquisition, Investigation, Writing—original draft, Writing—review and editing

## Author ORCIDs

David M Raizen, http://orcid.org/0000-0001-5935-0476

## Additional files

### Supplementary files

• Supplementary file 1. List of neurons in which *dsmr-1* is either expressed or not expressed.

• Supplementary file 2. Sequences of oligonucleotides used for sequencing and cloning.

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
