## [Decision Letter]

Thank you for submitting your article "The RFamide receptor DMSR-1 regulates stress-induced sleep in *C. elegans*" for consideration by *eLife*. Your article has been reviewed by two peer reviewers, and the evaluation has been overseen by Eve Marder as the Senior Editor and Reviewing Editor. The following individuals involved in review of your submission have agreed to reveal their identity: Patsy Dickinson (Reviewer #1).

The reviewers have discussed the reviews with one another and the Reviewing Editor has drafted this decision to help you prepare a revised submission.

Please note that both reviewers felt the paper would have substantial impact on the field, but also felt that editorial work would strengthen its impact. In this case, I believe it will be helpful for you to see the initial reviews because they are in essential agreement, and focus on queries that you should answer to make the manuscript more broadly accessible and useful.

*Reviewer #1:*

This paper provides convincing evidence that the gene dmsr-1 encodes a receptor (DMSR-1) that is activated by FLP-13 peptides, and is important in controlling 'stress-induced sleep', that is, quiescence induced by the effects heat stress or of EGF (presumably acting on the ALA neuron). Using a mutagenesis screen, the authors found that a number of mutants that failed to respond to overexpression of *flp-13*, and therefore presumably over-stimulation by the FLP-13 peptides had a variety of mutations in the *dmsr-1* gene, leading to a failed DMSR-1 receptor. They went on to show that the DMSR-1 receptor binds to the FLP-13 peptides, that it is required for stress-induced sleep, that the *dmsr-1* gene is expressed in the nervous system (in a number of specific neurons), and finally that the likely mechanism for the effects of DMSR-1 activation on stress-induced sleep are likely mediated by an inhibition of wake-promoting neurons rather than by an activation of sleep-inducing neurons.

The writing is clear, and it appears to me that the data are appropriately interpreted and make a strong case for their interpretation. Moreover, the manuscript should be of interest to a reasonably wide range of neuroscientists. The ability to manipulate the genetic makeup of *C. elegans*, together with the existing knowledge of the nervous system, make this a system that works very well for this study.

*Reviewer #2:*

This manuscript identifies the GPCR DMSR-1 as a novel regulator of stress-induced sleep through an unbiased forward genetic screen for genes that suppress the effects of a sleep-promoting neuropeptide. Complex neural circuits regulate sleep, and investigating sleep regulation in the nematode's relatively simple nervous system has potential to identify defined circuits regulating this process, and how they are modulated by environmental change. These findings build upon previous work identifying the ALA neurons as mediating stress-induced changes in sleep by release of FLP-13 neuropeptide. The identification of DMSR-1 as a target of FLP-13 provides a system for examining neurohormonal modulation of sleep in response to environmental or physiological changes. The experiments are technically sound and will have broad impact on the sleep community. Below I list a number of comments, mostly minor, that I believe would increase the accessibility and impact of this manuscript.

1) Because this journal is geared towards a generalist audience, greater description of the assays and interpretations of results would be helpful.

2) It is not clear why heat was selected as a stressor to characterize SIS in DMSR mutants while UV was used in the HisCL experiments. Using 35 degrees? What about other stressors? It would be very interesting to know if the role of DMSR-1 is generalizable across difference stressors.

3) The impacted would be aided by localizing DMSR-1 function. Speculate on what other. Is it at least possible to separate head and tail populations?

4) In Figure 7, the double mutant effect almost looks additive, which is somewhat surprising. This experiment may benefit from a more detailed analysis e.g. a number of different UV intensities or heat gradients.

---

## [Author Response]

*Please note that both reviewers felt the paper would have substantial impact on the field, but also felt that editorial work would strengthen its impact. In this case, I believe it will be helpful for you to see the initial reviews because they are in essential agreement, and focus on queries that you should answer to make the manuscript more broadly accessible and useful.*

We thank the editor and the reviewers for spending the time to closely read the manuscript and for their constructive comments, which served to strengthen the manuscript. We performed additional experiments in an attempt to localize the function of *dmsr-1* within the nervous system. These new data are now shown in Figure 5—figure supplement 2. We made editorial changes to the manuscript, as guided by the two reviewers’ comments.

*Reviewer #2:*

*[…] Below I list a number of comments, mostly minor, that I believe would increase the accessibility and impact of this manuscript.*

1) Because this journal is geared towards a generalist audience, greater description of the assays and interpretations of results would be helpful.

We added greater details of the methods, the results, and our interpretation of the results.

*2) It is not clear why heat was selected as a stressor to characterize SIS in DMSR mutants while UV was used in the HisCl experiments.*

We currently have under review a manuscript showing that short wavelength ultraviolet light triggers sleep, and that this sleep is regulated by ALA and FLP-13, similar to the regulation of SIS by heat and other stressors described in Hill et al. and Nelson et al., 2014. Because UV induces quiescence over a longer time course, we have found that it is more sensitive for detecting the differences found in this figure. This rationale has been clarified in the text. In addition, we now introduce the use of UV at an earlier point in the paper, now in Figure 3—figure supplement 1.

*Using 35 degrees? What about other stressors? It would be very interesting to know if the role of DMSR-1 is generalizable across difference stressors.*

SIS can be triggered at other temperatures – in general, the higher the temperature, the greater the subsequent behavioral response. Please see Figure 3 in Nelson et al., Current Biology 2014. To test whether *dmsr-1* is required for SIS after exposure to temperatures other than 35 degrees C, we performed an additional experiment in which we exposed cohorts of animals to 30 minutes of heat at temperatures ranging from 29 degrees Celsius to 37 degrees C. These new data are now shown in Figure 3—figure supplement 1.

To determine whether the role in *dmsr-1* can be generalized to other stressors, we performed an additional experiment in which we tested the effect of the mutant on sleep following exposure to ultraviolet C radiation. These new data are shown in Figure 3—figure supplement 1. Hence, indeed the role of *dmsr-1* in SIS appears to be generalizable to different degrees of stress and to different types of stress.

*3) The impacted would be aided by localizing DMSR-1 function. Speculate on what other. Is it at least possible to separate head and tail populations?*

We performed two cell-specific (AIY and PHA) rescue experiments but were unable to fully rescue the *dmsr-1* quiescence defect. With AIY-specific expression, we did find a small but statistically significant rescue of the *flp-13* overexpression-induced body movement quiescence, but no rescue of the *flp-13* overexpression-induced feeding quiescence and no rescue of the body movement or feeding movement quiescence during SIS. While it is certainly possible that *dmsr-1* function is required in a neuron type that we did not yet test, the explanation we favor is that *dmsr-1* function is simultaneously required in several neurons. We discuss these results in the context of our model of a distributed wake-promoting system.

*4) In Figure 7, the double mutant effect almost looks additive, which is somewhat surprising. This experiment may benefit from a more detailed analysis e.g. a number of different UV intensities or heat gradients.*

We agree that the double mutant has a stronger effect than either of the single mutants. We explain this finding by proposing that FLP-13 peptides are acting on other receptors and that the DMSR-1 receptor responds to other peptides. Performing a full dose response analysis of the stressors is unlikely to illuminate the interpretation of this double mutant phenotype.